# Assessing the New 2020 ESGO/ESTRO/ESP Endometrial Cancer Risk Molecular Categorization System for Predicting Survival and Recurrence

**DOI:** 10.3390/cancers16050965

**Published:** 2024-02-27

**Authors:** Yung-Taek Ouh, Yoonji Oh, Jinwon Joo, Joo Hyun Woo, Hye Jin Han, Hyun Woong Cho, Jae Kwan Lee, Yikyeong Chun, Myoung-nam Lim, Jin Hwa Hong

**Affiliations:** 1Department of Obstetrics and Gynecology, Ansan Hospital, Korea University College of Medicine, Ansan 15355, Republic of Korea; oytjjang@gmail.com; 2Department of Obstetrics and Gynecology, Guro Hospital, Korea University College of Medicine, Seoul 02841, Republic of Korea; dbsel91@gmail.com (Y.O.); jjw2311@gmail.com (J.J.); woojoohyun@gmail.com (J.H.W.); anasta930408@gmail.com (H.J.H.); limpcho82@gmail.com (H.W.C.); jklee38@gmail.com (J.K.L.); 3Department of Pathology, Guro Hospital, Korea University College of Medicine, Seoul 02841, Republic of Korea; ykcmd@naver.com; 4Biomedical Research Institute, Kangwon National University Hospital, Chuncheon 24289, Republic of Korea; lmn99054@kangwon.ac.kr

**Keywords:** endometrial cancer, molecular classification, mismatch repair, polymerase epsilon, p53

## Abstract

**Simple Summary:**

This study explores the use of the 2020 ESGO/ESTRO/ESP molecular classification system in predicting survival and recurrence rates in patients with endometrial cancer. It aims to assess the effectiveness of this new classification by comparing it with previous ESMO guidelines. By incorporating both clinicopathological and molecular factors into the risk assessment, this research seeks to offer a more precise method for categorizing patients, potentially leading to more tailored and effective treatment strategies. The findings could significantly influence future approaches to managing endometrial cancer, emphasizing the importance of molecular risk categorization in improving patient outcomes.

**Abstract:**

This study aimed to evaluate the efficacy of the 2020 European Society of Gynecological Oncology/European Society for Radiotherapy and Oncology/European Society of Pathology (ESGO/ESTRO/ESP) guidelines for endometrial cancer (EC). Additionally, a novel risk category incorporating clinicopathological and molecular factors was introduced. The predictive value of this new category for recurrence and survival in Korean patients with EC was assessed, and comparisons were made with the 2013 and 2016 European Society of Medical Oncology (ESMO) risk classifications. Patients with EC were categorized into the POLE-mutated (POLEmut), mismatch repair-deficient (MMRd), p53-aberrant (P53abn), and nonspecific molecular profile (NSMP) subtypes. Recurrence, survival, and adjuvant therapy were assessed according to each classification. Notably, patients with the POLEmut subtype showed no relapse, while patients with the P53abn subtype exhibited higher recurrence (31.8%) and mortality rates (31.8%). Regarding adjuvant therapy, 33.3% of low-risk patients were overtreated according to the 2020 ESGO/ESTRO/ESP guidelines. Overall and progression-free survival differed significantly across molecular classifications, with the POLEmut subtype showing the best and the P53abn subtype showing the worst outcomes. The 2020 ESGO molecular classification system demonstrated practical utility and significantly influenced survival outcomes. Immunohistochemistry for TP53 and MMR, along with POLE sequencing, facilitated substantial patient reclassification, underscoring the clinical relevance of molecular risk categories in EC management.

## 1. Introduction

Globally, endometrial cancer (EC) has exhibited a substantial annual increase in incidence (approximately 0.69%) from 1990 to 2019, and the mortality trend continues to rise [1]. The yearly rate of EC in Republic of Korea saw growth from 1999 to 2009, with an annual percent change (APC) of 7.24%. This growth slowed between 2009 and 2014, with an APC of 3.29%, indicating a stabilization in the incidence rate. However, between 2014 and 2017, the incidence rate surged rapidly, with an APC of 8.74%, reflecting a significant acceleration in new cases of EC during this period [2]. In Korea, the age-standardized rate (ASR) of EC was 2.4 in 1999, demonstrating a significant increase to an ASR of 7.0 by 2017. According to the cancer registry statistics in Korea, as of 2021, endometrial cancer is the eighth most common cancer, with an incidence rate of 14.6% [3]. The APC of the overall EC ASR increased by 5.0%, while the APC of the overall cervical cancer ASR decreased by 4.0% between 1999 and 2020. This resulted in EC becoming more prevalent than cervical cancer for the first time in 2019, and this trend continues [4]. The management of patients with advanced and recurrent EC poses a significant therapeutic challenge, particularly given the poor 5-year overall survival rate of only 17% for individuals with distant-stage EC [5]. In contrast, over 80% of patients with early-stage EC have a good prognosis, with a 95% overall survival rate at 5 years [5]. Nonetheless, within the latter group, approximately 20% have one or more high-risk characteristics that increase their vulnerability to cancer-related death. Determining which subgroup of patients with early-stage EC is low-risk (i.e., has a recurrence risk of less than 5%) and may therefore be effectively treated with surgery alone is a critical diagnostic challenge. This differentiation is important when comparing high-risk patients who require adjuvant therapy.

In 2013, The Cancer Genome Atlas (TCGA) research network conducted a comprehensive genomic analysis of 373 cases of EC using sequencing and array-based technologies [6]. These investigations led to the development of the ProMisE (Proactive Molecular risk classifier for EC) molecular classification, which identifies four distinct molecular categories of EC, each associated with unique prognostic implications [7]. These categories include the POLE-mutated group (POLEmut), the mismatch repair-deficient group (MMRd), the p53-aberrant group (P53abn) characterized by a “high copy number”, and the p53-wild-type group (p53-wt) or “non-specific molecular profile” (NSMP).

Notably, the ProMisE classification system has proven to be a valuable supplement to the 2013 ESMO classification system. When evaluating primary oncological outcomes such as overall survival (OS), disease-free survival, and progression-free survival (PFS), the use of the ProMisE system alone appears to be as effective as or potentially superior to the ESMO 2013 risk classification. This is especially true when considering postoperative parameters [8,9]. Moreover, the updated 2020 ESTRO/ESGO/ESP guidelines have successfully integrated molecular classification into management algorithms [10]. This integration has led to the modification of risk groups, subsequently impacting the medical and surgical management of ECs. The objective of applying a precise prognostic classification based on molecular subtypes is to reduce iatrogenic morbidity. This reduction is achieved by minimizing unnecessary adjuvant therapies, aligning with the 2020 ESGO guidelines, and efficiently reserving these treatments for high-risk patients.

We aimed to assess the application of the 2020 ESGO/ESTRO/ESP molecular risk groups in a substantial cohort of patients with EC. Additionally, we conducted a comparative analysis with the clinicopathologic-only risk groups from the ESMO 2016 risk classification.

## 2. Materials and Methods

### 2.1. Study Population

All patients underwent surgery for EC at Guro Hospital, Korea University, between January 2012 and December 2021. Patients in whom the tissue was unsatisfactory due to alterations occurring during preservation or poor DNA quality were excluded. Furthermore, patients who were diagnosed with endometrial cancer through tissue biopsy or similar methods but did not undergo a hysterectomy were excluded.

The adjuvant therapy for the patients involved in this study was determined based on either the ESMO 2013 or ESMO 2016 guidelines, with the specific treatment tailored according to the individual circumstances and clinical judgment of the treating physicians.

The following information was collected: age, parity, stage, histological type, tumor grade, tumor size, depth of myometrial invasion, lymphovascular space invasion, cervical involvement, lymph node involvement, adjuvant therapy, and survival.

### 2.2. Molecular Classification

DNA was isolated from formalin-fixed, paraffin-embedded tumor tissues using a QIAamp DNA FFPE Mini Kit (QIAGEN, Germantown, MD, USA). POLE exonuclease domain mutation screening of hotspots in exons 9 (c.857C>G, p.P286R; c.890C>T, p.S297F), 13 (c.1231G>C, p.V411L), and 14 (c.1366G>C, p.A456P) was performed by Sanger sequencing, as previously described [11]. Mutation analysis was conducted using Variant Reporter software ver. 2.0 (Applied Biosystems, Thermo Fisher Scientific, Waltham, MA, USA) and manual inspection.

The dMMR/MSI-H status was determined by immunohistochemical (IHC) staining and/or MSI analysis. Tissue microarrays were constructed from paraffin-embedded blocks from 123 patients with ECs. IHC staining was conducted using a BOND-III automated staining system (Leica, Wetzlar, Germany) with the following antibodies: MLH1 (1:100, ES05, Novocastra (Vision Bio Systems Europe, Newcastle upon Tyne, UK)), MSH2 (1:400, G219-1129, Novocastra), PMS2 (1:100, MRQ-28, Cell Marque), and MSH6 (1:200, 44, Cell Marque). The MMR immunostaining results were classified as either retained or lost nuclear expression. Retained nuclear expression was considered MMR-proficient, whereas the loss of nuclear expression was considered MMR-deficient. MSI analysis was performed using either a Bethesda or Pentaplex panel. Normal and tumor allele patterns were compared for each marker. Two or more positive markers were classified as MSI-H, and one or zero as MSS.

The IHC staining for p53 (1:200, DO-7, Novocastra) was performed. The p53 staining was considered abnormal or mutant if more than 80% of the tumor cells exhibited diffuse strong nuclear staining, unequivocal cytoplasmic staining, or no nuclear or cytoplasmic staining. Cases were classified as normal/wild-type if any degree of nondiffuse nuclear staining (<80%) was present in the tumor cells.

Patients with pathogenic POLE mutations (POLEmut) were categorized for molecular classification. The remaining endometrial carcinomas (ECs) were then classified according to their mismatch repair (MMR) status, identifying patients whose tumors exhibited loss of one or more MMR proteins (MMRd). Finally, p53 status was employed to differentiate between abnormal (P53abn) and normal (NSMP/p53wt) staining patterns.

### 2.3. Assessment of the 2020 ESGO/ESTRO/ESP Classification

All patients were reclassified according to the new ESGO/ESTRO/ESP classification. Subsequently, the initially assessed risk during prospective management was compared with the new risk grouping. A comparison between adjuvant therapy and the surgical approach was conducted to evaluate the impact of the revised 2020 ESGO/ESTRO/ESP guidelines.

For cases with sufficient data, ESMO 2016 [12] and 2020 ESGO/ESTRO/ESP (molecular) risk groups [10] were assigned, and the suggested therapeutic changes were compared to ESMO 2013 guidelines [13]. Patient survival was analyzed based on molecular groups, prognostic risk groups, and histological features.

### 2.4. Statistical Analyses

We utilized the chi-squared test or Fisher’s exact test to explore univariate relationships between binary and categorical variable centers. The association between the outcome and ProMisE subtype was investigated and depicted through Kaplan–Meier curves along with a log-rank test. Statistical significance was considered at a predetermined level to be statistically significant.

## 3. Results

### 3.1. Characteristics of the Population

A total of 136 patients were enrolled, and molecular classification was performed for all participants (Table 1). The age at diagnosis was distributed as follows: 52.2% for individuals aged 60 and older, 46.3% for those between 40 and 60 years old, and 1.5% for individuals under 40 years old. Parity was reported as 2 in 67 patients, constituting 49.3% of the population, and 3 or more in 20 patients, accounting for 14.7% of the total population. Regarding histological type, 122 patients (89.7%) exhibited the endometrioid type, while 14 patients (10.3%) presented with the non-endometrioid type. According to the FIGO 2009 staging system for EC, IA was the most prevalent stage (91 patients, 66.9%), followed by IB with 23 patients (16.9%), II with 5 patients (3.7%), IIIA with 8 patients (5.9%), IIIC1 with 4 patients (2.9%), IIIC2 with 1 patient (0.7%), IVA with 1 patient (0.7%), and IVB with 3 patients (2.2%). In terms of pathological grade, G1 was the most common grade (83 patients, 61.0%), followed by G2 (30 patients, 22.1%) and G3 (23 patients, 16.9%). Myometrial invasion was absent in 43 patients (31.6%), while muscular invasion was observed in 36 patients (26.5%).

Lymphovascular space invasion was observed in 33 patients (24.3%), with low uterine segment involvement in 21 patients (15.4%) and cervical involvement in 9 patients (6.6%). Lymph node metastasis occurred in 7 patients (5.1%), and lymph node dissection was not performed in 22 patients (16.2%). Postoperative adjuvant therapy included brachytherapy for 32 patients (23.5%), external beam radiation therapy (EBRT) for 10 patients (7.4%), chemotherapy for 14 patients (10.3%), and concomitant chemoradiotherapy (CCRT) for 11 patients (8.1%). Recurrence was reported in 12 patients (8.8%), and death was observed in 10 patients (7.4%).

### 3.2. Molecular Classification

In terms of molecular classification, NSMP was the most common (80/136, 58.8%), followed by MMRd/MSI-H (25/136, 18.4%), P53abn (22/136, 16.2%), and POLEm (9/136, 6.6%) (Table 1). All nine patients with tumors classified as POLEmut were of the endometrioid type, and eight (88.99%) were determined to have a grade of 1–2. In contrast, of the 22 patients with tumors classified as P53abn, 14 (63.6%) had non-endometrioid-type lesions, and 17 (77.3%) had high-grade lesions. All patients with POLEmut tumors had a depth of myometrial invasion of 50% or less, whereas 12 (54.4%) of the patients with P53abn tumors had a depth of myometrial invasion of 50% or more. Among patients with POLEmut tumors, none experienced relapses, whereas in those with P53abn tumors, seven (31.8%) recurrences and seven (31.8%) deaths were observed.

### 3.3. Adjuvant Therapy of the Patients

According to the 2020 ESGO/ESTRO/ESP risk classification, 78 patients in the low-risk group were recommended not to undergo adjuvant therapy. However, in our dataset, 26 patients (33.3%) received unnecessary treatment; 24 (30.8%) underwent brachytherapy, and 2 (2.6%) received EBRT (Table 2).

In the intermediate-risk group, where observation or brachytherapy was recommended according to the 2020 ESGO/ESTRO/ESP guidelines, 16 patients were classified. Among them, seven patients (43.8%) received overtreatment—four received EBRT, one received concurrent chemoradiotherapy (CCRT), and two received chemotherapy.

Fourteen patients were categorized in the high-intermediate group based on the 2020 ESGO/ESTRO/ESP risk classification, necessitating treatment beyond adjuvant brachytherapy. However, nine of these patients did not undergo adjuvant therapy. Among them, two opted for brachytherapy, one for EBRT, and two for chemotherapy.

Of the 23 patients classified in the high-risk group, for whom EBRT with concurrent and adjuvant chemotherapy or chemotherapy alone was recommended, 3 did not receive adequate adjuvant treatment. The first patient had stage IA, grade 3 endometriosis, initially deemed intermediate risk according to the ESMO 2013 risk classification. However, subsequent assessments in 2016 and 2020 identified her as high-intermediate and high-risk, respectively. Despite requiring EBRT, she only received brachytherapy and experienced no recurrence. The other two patients were initially classified as low risk in 2013 but were later categorized as high-intermediate risk in 2016 due to positive LVSI, and as high-risk in 2020. Despite being in the high-risk group, none of these patients received adjuvant therapy. One patient relapsed and died, while the others remained in complete remission without relapse.

Five patients were classified into the advanced/metastatic group based on the 2020 ESGO/ESTRO/ESP risk classification. Two received chemotherapy, two underwent CCRT, and one opted for brachytherapy.

### 3.4. Shift in Risk Groups between ESMO 2013 and ESMO 2016 Risk Classification

Among the 86 patients initially classified as low-risk according to the ESMO 2013 classification, 9 experienced an increase in their risk level to high/intermediate (Figure 1). Out of the 21 patients initially classified as intermediate from the ESMO 2013 classification, 1 was reclassified as low-risk, and 8 were reclassified as high/intermediate. Additionally, among the 29 patients initially classified as high-risk from the ESMO 2013 classification, 6 were subsequently reclassified as advanced/metastatic.

### 3.5. Shift in Risk Groups between 2016 ESMO and 2020 ESGO

Out of the 136 patients, 15 experienced group realignments based on the 2020 ESGO criteria in comparison to the ESMO 2016 criteria (Table 3). Among these 15 patients who underwent group shifting, 6 were reclassified into a higher-risk group, and 9 were moved to a lower-risk group compared to the ESMO 2016 classification (Figure 1). Of the 78 patients initially classified as low-risk according to the ESMO 2016 criteria, 1 was reclassified as high-risk due to the presence of P53abn tumors. Additionally, two were upgraded to the intermediate-risk group because of detected p53 abnormalities. Among the 17 patients initially classified as high-intermediate risk by the ESMO 2016 criteria, 4 (23.5%) were shifted to the high-risk group following the confirmation of P53abn tumors. Within the group of 23 patients initially classified as high-risk according to the ESMO 2016 criteria, 6 patients were downgraded. One patient, initially classified as high-risk due to cervical involvement, was reclassified as low-risk following the 2020 ESGO/ESTRO/ESP guidelines, as POLEmut tumors were detected. Two patients, initially considered high-risk based on the ESMO 2016 guidelines due to their serous histological type, were downgraded to intermediate-risk because they exhibited no myometrial invasion and were classified into the P53abn group. Among the remaining three patients, two had cervical involvement, and one had deep myometrial invasion. According to the ESMO 2016 guidelines, they would be considered high-risk, but they were downgraded to high-intermediate risk because they fell into the MMRd/MSI-H group.

### 3.6. Prognosis

The classification of risk groups according to the 2020 ESGO criteria revealed statistically significant differences in both OS and PFS (*p* < 0.0001) (Figure 2). Moreover, irrespective of the FIGO 2009 staging system, molecular classification alone exhibited significant variations in overall survival and PFS. The POLEmut group demonstrated the most favorable prognosis, while the P53abn group had the least favorable prognosis (*p* < 0.0001) (Figure 3). Among the nine patients in the POLEmut group, there was one patient each at stages IB, II, and III, and one patient with lymphovascular space invasion (LVSI). Despite these clinical variables, not a single patient experienced recurrence.

Five patients were reclassified into the high-risk group according to the 2020 ESGO/ESTRO/ESP risk classification, as opposed to the ESMO 2016 risk classification. Among them, four patients did not experience recurrence or death, while one patient had recurrence with liver metastases and subsequently succumbed. Conversely, two patients were reclassified into the low-risk group according to the 2020 ESGO/ESTRO/ESP risk classification, and notably, neither of them experienced recurrence nor death. Furthermore, three patients were downgraded to the intermediate-risk group in 2020. One of them, despite receiving adjuvant therapy, experienced recurrence approximately one year later.

## 4. Discussion

We assessed the feasibility of using the 2020 ESGO/ESTRO/ESP predictive risk groups in an unselected population of patients with EC. Notably, the 2020 ESGO/ESTRO/ESP molecular classification resulted in significant shifts in risk groups, thereby influencing treatment decisions. Deviations in adjuvant therapy were observed, with some patients being overtreated, while others did not receive the recommended treatments based on the new classification. Shifts between the ESMO 2013, ESMO 2016, and 2020 ESGO/ESTRO/ESP risk classifications were evident, reflecting the dynamic nature of risk assessment. Prognostically, the 2020 ESGO/ESTRO/ESP risk groups demonstrated significant differences in OS and PFS, emphasizing the clinical relevance of molecular classification independent of FIGO staging. Our findings highlighted the potential impact of adopting the 2020 ESGO/ESTRO/ESP guidelines on patient management and outcomes.

Advancements in EC management have been hindered by inconsistent histological classifications and grading systems, leading to varied approaches in managing EC. The need for change stems from various factors: pathologists seeking a more reliable classification system, clinicians observing patients undergoing overly aggressive or insufficient treatment based on unreliable criteria, and researchers grappling with the interpretation of data from heterogeneously grouped tumors. Progress in managing EC has encountered obstacles due to acknowledged inconsistencies in the risk classification of histology and grade [14,15]. The existing risk stratification system inadequately represents the tumor biology and behavior of the disease, resulting in substantial variations in management, as illustrated in this study. The observed variability in clinical practice signifies uncertainty regarding the optimal treatment for individual patients—a situation that is vulnerable but can be corrected through enhanced, more accurate, and reproducible risk stratification methods. The application of molecular prognostic groupings in the 2020 ESGO/ESTRO/ESP risk classification resulted in notable variations across all survival outcomes, confirming the effectiveness of the updated risk categorization. The most recent modification, which introduced LVSI in 2016 [16,17], demonstrated comparable outcomes in stratifying risk groups. The integration of 2020 molecular risk groups into the cohort revealed several crucial aspects [1].

The ProMisE subtype patterns identified in our study were consistent with the distributions reported in the literature [1,8,9,18]. In all patients, the prevailing molecular subtype was NSMP, accounting for 58.8%, followed by MMRd/MSI-H at 18.4%, P53abn at 16.2%, and POLEmut at 6.6%. Within the POLEmut group, comprising nine patients, five were concurrently categorized as low-risk according to the ESMO 2016 risk classification, resulting in no alteration in their risk group status. One patient with histological stage IIIC1 remained outside the high-risk category in both classifications. Notably, three patients transitioned from the intermediate-, high-intermediate-, and high-risk categories to the low-risk group in the 2020 ESGO/ESTRO/ESP risk classification. Of particular interest is one patient initially classified in the high-risk group with histological stage II who managed to survive without any recurrence, thus moving to the low-risk category. In an assessment of previous studies, including the new risk classifications, it was observed that 8% of patients initially categorized as high-risk were assigned to the low-risk group due to the presence of POLEmut tumors [19]. This finding is noteworthy, as the POLEmut subtype has been described as uncommon in previous investigations [20,21].

In this study, the association of ProMisE molecular subtypes with clinical outcomes, such as PFS and OS, was demonstrated, confirming the significance of molecular classification in prognosis. The research also highlighted overlooked therapeutic opportunities linked to molecular subtypes. For instance, P53abn subtypes are indicative of chemotherapy responsiveness, whereas MMRd subtypes show the potential for increased radiotherapy effectiveness [22,23,24]. The prognostic value of ProMisE molecular classification has been well established for high-risk EC [25]. Different molecular subtypes, such as P53abn and POLEmut, exhibited varied responses to adjuvant chemotherapy and radiotherapy, influencing patient outcomes. Currently, adjuvant treatment recommendations are made based on a mix of clinical (such as age) and pathological (like FIGO stage, tumor type and grade, and clear LVSI presence) factors that categorize a patient’s risk as low, intermediate, or high [26]. The integration of additional molecular data into this stratification system is yet to be defined. Specifically, the P53abn subtypes significantly benefited from combined adjuvant chemotherapy and radiotherapy, whereas the POLEmut subgroup showed outstanding survival rates regardless of adjuvant therapy. The PORTEC-3 study, focusing on high-risk endometrial cancer, compared the outcomes of chemoradiation versus radiation alone [25,27]. It found that the 5-year recurrence-free survival (RFS) for patients with P53abn endometrial cancer was significantly higher when treated with both chemotherapy and radiotherapy (59%) compared to radiotherapy alone (36%). In contrast, for POLE-mutated endometrial cancer, the RFS was nearly perfect at 100% versus 97%, showing no significant difference. Our study emphasized the critical role of molecular classification in guiding treatment decisions and improving risk stratification in EC management.

Patients in the POLEmut subgroup were offered a chance to reduce adjuvant therapy, thereby avoiding treatment-related toxicities. Tumors characterized by POLE-proofreading mutations are associated with considerably better prognoses even in higher stages compared to other subgroups, with notable enhancements in overall survival, disease-specific survival, and progression-free survival outcomes linked to POLE mutations. This is particularly significant in grade 3 endometrioid endometrial cancer (EC), where POLE mutations imply a reduced likelihood of disease progression and metastasis, including diminished myometrial invasion [28]. The presence of POLE mutations elicits an antitumor immune response, further contributing to the improved prognosis of these patients. This finding is supported by both individual patient data meta-analyses and retrospective studies, including one with 26 patients with high-grade POLEmut EC who experienced no negative effects without adjuvant treatment. However, ongoing trials are needed to confirm the safety of this reduced treatment approach for POLEmut EC, as well as for other molecular subtypes such as P53abn, MMRd, and NSMP EC [22,29]. Although the impact of molecular-directed treatments on outcomes remains uncertain, adherence to the 2020 ESGO/ESTRO/ESP guidelines and knowledge of ProMisE subtypes could potentially alter adjuvant therapy. In our retrospective analysis, we observed some degree of treatment deviation in our cohort. Specifically, 26 of the 78 patients (33.3%) in the low-risk group and 7 of the 17 patients (41.2%) in the intermediate-risk group appeared to have received more treatment than is recommended in the 2020 ESGO/ESTRO/ESP guidelines. In contrast, 9 out of 14 patients (64.3%) in the high-intermediate group, and 3 out of 22 patients (13.6%) in the high-risk group received less treatment than might have been optimal. This highlights the potential for both overtreatment and undertreatment in the different risk stratifications of the cohort.

The application of the 2020 ESGO/ESTRO/ESP molecular risk groups, in conjunction with the clinicopathologic-only risk groups from the ESMO 2016, offers a comparative perspective, enhancing our understanding of risk stratification in EC. Through the utilization of advanced techniques, such as next-generation sequencing and comprehensive molecular profiling, we provide a detailed molecular characterization of ECs. Additionally, we effectively correlate molecular subtypes with clinical outcomes, such as OS and PFS, emphasizing the prognostic significance of molecular classification. Finally, we shed light on deviations in adjuvant therapy, highlighting areas of overtreatment and undertreatment in different risk categories. Nevertheless, our study has some limitations owing to its retrospective design, and it may be subject to biases inherent in such designs, including selection and information biases. Secondly, given the evolving nature of molecular classification, there is a risk of misclassification as new information emerges.

The integration of molecular classification into pathological reporting, as initiated by the WHO Health Organization guidelines in September 2020, offers a pathway for future change. This integration, coupled with the new FIGO staging system for EC introduced in 2023 [30], which emphasizes molecular classification, can reduce practice variations and healthcare disparities. This also paves the way for validating the predictive value of molecular subtypes in treatment decisions. Gynecologic oncologists must familiarize themselves with the updated system and understand the factors that influence the modification of risk categorization before its implementation.

## 5. Conclusions

The application of the 2020 ESGO/ESTRO/ESP guidelines highlights their feasibility and reveals significant survival disparities. The identification of molecular risk groups in 2020 signifies a fundamental shift in EC assessment and management. Compared to the 2016 clinicopathological risk groups, the 2020 ESGO classification resulted in considerable patient reclassifications, especially for those with P53 abnormalities. This suggests the potential for the streamlined adoption of the new classification, offering efficiency benefits.

## Figures and Tables

**Figure 1 cancers-16-00965-f001:**
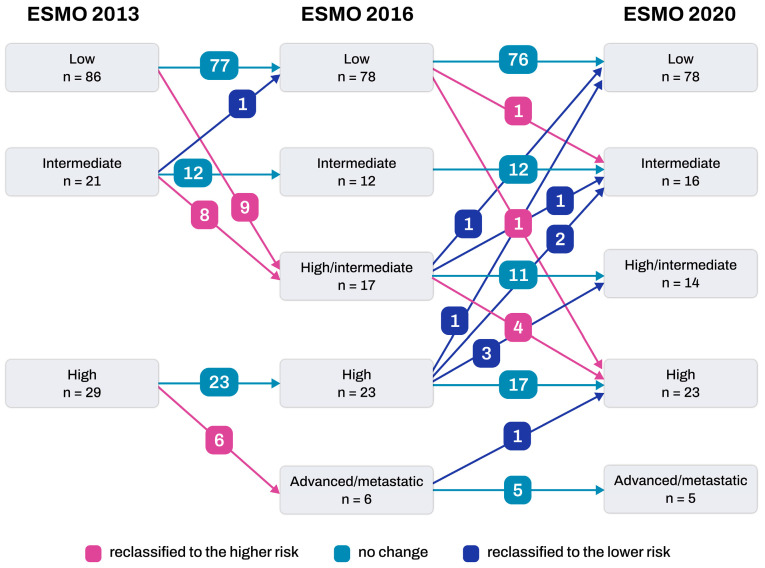
Changes in patient populations by three different classifications.

**Figure 2 cancers-16-00965-f002:**
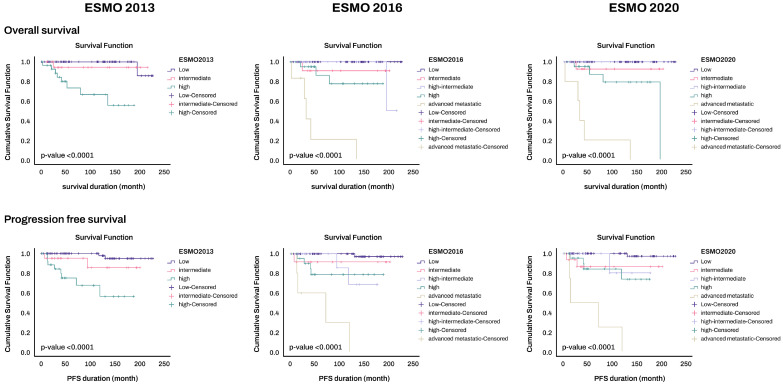
Overall survival and progression-free survival according to ESMO 2013, ESMO 2016, and ESGO 2020 risk classification systems.

**Figure 3 cancers-16-00965-f003:**
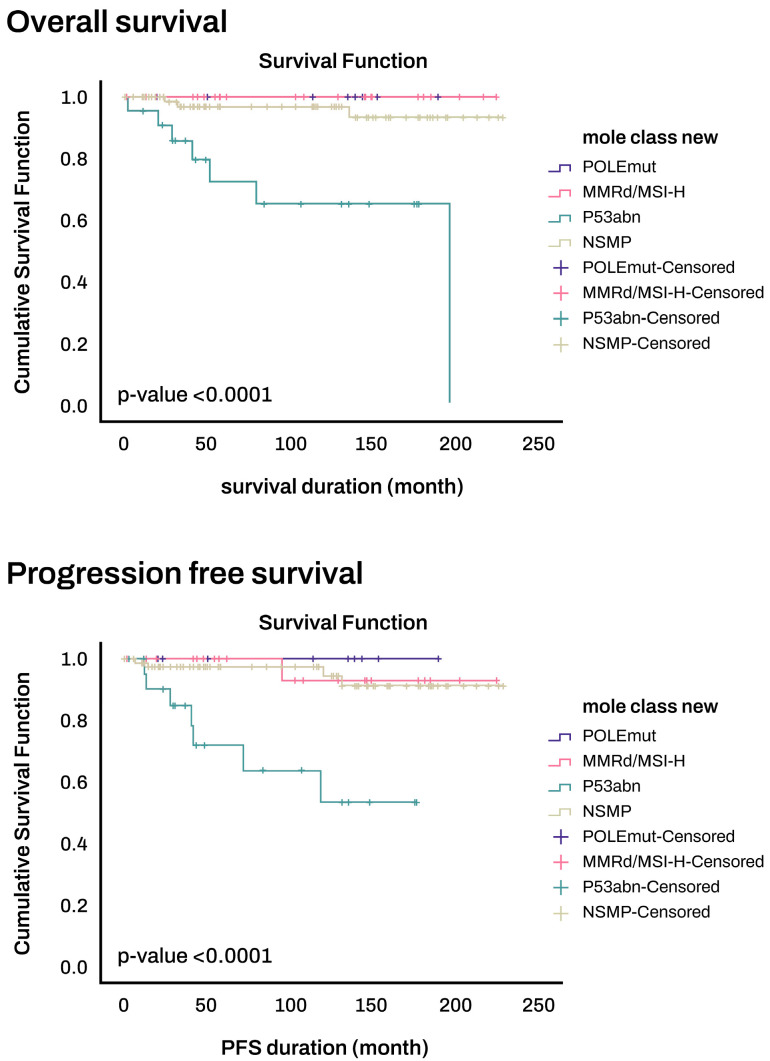
Overall survival and progression-free survival stratified by molecular classification.

**Table 1 cancers-16-00965-t001:** Baseline characteristics of the study population.

Variables	N (%)	Molecular Classification of the Tumors			
		POLEmut	MMRd/MSI-H	P53abn	NSMP
**Number (%)**	136 (100%)	9 (6.6%)	25 (18.4%)	22 (16.2%)	80 (58.8%)
**Age (years)**					
<40	2 (1.5%)	0 (0.0%)	0 (0.0%)	0 (0.0%)	2 (2.5%)
40–59	63 (46.3%)	6 (66.7%)	17 (68.0%)	6 (27.3%)	34 (42.5%)
>60	71 (52.2%)	3 (33.3%)	8 (32.0%)	16 (72.7%)	44 (55.0%)
**Parity**					
0	24 (17.6%)	2 (22.2%)	2 (8.0%)	3 (13.0%)	17 (21.3%)
1	25 (18.4%)	3 (33.3%)	6 (24.0%)	5 (22.7%)	11 (3.8%)
2	67 (49.3%)	4 (44.4%)	14 (56.0%)	9 (40.9%)	40 (50.0%)
>3	20 (14.7%)	0 (0.0%)	3 (12.0%)	5 (22.7%)	12 (15.0%)
**Histology**					
Endometrioid	122 (89.7%)	9 (100.0%)	25 (100.0%)	8 (36.4%)	80 (100.0%)
Non-Endometrioid	14 (10.3%)	0 (0.0%)	0 (0.0%)	14 (63.6%)	0 (0.0%)
**FIGO stage**					
IA	91 (66.9%)	6 (66.7%)	17 (68.0%)	7 (31.8%)	61 (76.3%)
IB	23 (16.9%)	1 (11.1%)	4 (16.0%)	5 (22.7%)	13 (16.3%)
II	5 (3.7%)	1 (11.1%)	2 (8.0%)	2 (9.1%)	0 (0.0%)
III	13 (9.6%)	1 (11.1%)	2 (8.0%)	6 (27.3%)	4 (5.0%)
IVA	1 (0.7%)	0 (0.0%)	0 (0.0%)	0 (0.0%)	1 (1.3%)
IVB	3 (2.2%)	0 (0.0%)	0 (0.0%)	2 (9.1%)	1 (1.3%)
**FIGO 2023 stage**					
IA1	33 (24.3%)	0 (0.0%)	7 (28.0%)	0 (0.0%)	26 (32.5%)
IA2	38 (27.9%)	0 (0.0%)	8 (32.0%)	0 (0.0%)	30 (37.5%)
IA3	0 (0.0%)	0 (0.0%)	0 (0.0%)	0 (0.0%)	0 (0.0%)
IAmPOLEmut	8 (5.9%)	8 (88.9%)	0 (0.0%)	0 (0.0%)	0 (0.0%)
IB	12 (8.8%)	0 (0.0%)	2 (8.0%)	0 (0.0%)	10 (12.5%)
IC	0 (0.0%)	0 (0.0%)	0 (0.0%)	0 (0.0%)	0 (0.0%)
IIA	2 (2.2%)	0 (0.0%)	2 (8.0%)	0 (0.0%)	0 (0.0%)
IIB	12 (8.8%)	0 (0.0%)	4 (16.0%)	0 (0.0%)	8 (10.0%)
IIC	0 (0.0%)	0 (0.0%)	0 (0.0%)	0 (0.0%)	0 (0.0%)
IICmp53abn	14 (10.3%)	0 (0.0%)	0 (0.0%)	14 (63.6%)	0 (0.0%)
III	8 (5.9%)	0 (0.0%)	0 (0.0%)	6 (27.3%)	2 (2.5%)
IIIC	5 (3.7%)	1 (11.1%)	2 (8.0%)	0 (0.0%)	2 (2.5%)
IVA	1 (0.7%)	0 (0.0%)	0 (0.0%)	0 (0.0%)	1 (1.3%)
IVB	3 (2.2%)	0 (0.0%)	0 (0.0%)	2 (9.1%)	1 (1.3%)
**Grade**					
Low	113 (83.1%)	8 (88.9%)	22 (88.0%)	5 (22.7%)	78(97.5%)
High	23 (16.9%)	1 (11.1%)	3 (12.0%)	17 (77.3%)	2 (2.5%)
**Tumor size**	3.91 ± 2.61	4.01 ± 2.59	3.39 ± 1.95	4.68 ± 3.22	3.85 ± 2.61
**Depth of MI**					
None	43 (31.6%)	5 (55.6%)	7 (28.0%)	3 (13.6%)	28 (35.0%)
<1/2	57 (41.9%)	4 (44.4%)	11 (44.0%)	7 (31.8%)	35 (43.8%)
>1/2	36 (26.5%)	0 (0.0%)	7 (28.0%)	12 (54.5%)	17 (21.3%)
**LVSI**					
Absent	103 (75.7%)	8 (88.9%)	17 (68.0%)	10 (45.5%)	68 (85.0%)
Present	33 (24.3%)	1 (11.11%)	8 (32.0%)	12 (54.5%)	12 (15.0%)
**Lymph node**					
Unknown	22 (16.2%)	3 (33.3%)	1 (4.0%)	6 (27.3%)	12 (15.0%)
None	107 (78.7%)	5 (55.6%)	22 (88.0%)	15 (68.2%)	65 (81.3%)
Yes	7 (5.1%)	1 (11.1%)	2 (8.0%)	1 (4.5%)	3 (3.8%)
Residual tumor	5 (3.7%)	0 (0.0%)	0 (0.0%)	3 (13.6%)	2 (2.5%)
**Adjuvant therapy**					
None	69 (50.7%)	5 (55.6%)	14 (56.0%)	4 (18.2%)	46 (57.5%)
Brachytherapy	32 (23.5%)	3 (33.3%)	6 (24.0%)	2 (9.1%)	21 (26.3%)
EBRT	10 (7.4%)	0 (0.0%)	2 (8.0%)	3 (13.6%)	5 (6.3%)
CCRT	11 (8.1%)	1 (11.1%)	1 (4.0%)	6 (27.3%)	3 (3.8%)
Others	14 (10.3%)	0 (0.0%)	2 (8.0%)	7 (31.8%)	5 (6.3%)
**Recurrence**					
Yes	12 (8.8%)	0 (0.0%)	1 (4.0%)	7 (31.8%)	4 (5.0%)
None	124 (91.2%)	9 (6.6%)	24 (96.0%)	15 (68.2%)	76 (95.0%)
**Survival**					
Alive	121 (89.0%)	9 (100.0%)	24 (96.0%)	15 (68.2%)	73 (91.3%)
Death	10 (7.4%)	0 (0.0%)	0 (0.0%)	7 (31.8%)	3 (3.8%)
Unknown	5 (3.7%)	0 (0.0%)	1 (4.0%)	0 (1.1%)	4 (5.0%)

LVSI, lymphovascular space invasion; EBRT, external beam radiotherapy; CCRT, concurrent chemoradiotherapy.

**Table 2 cancers-16-00965-t002:** Adjuvant therapy in the context of ESMO 2016 and ESGO 2020.

Variables	Total, N (%)	None, N (%)	Brachytherapy, N (%)	EBRT, N (%)	CCRT, N (%)	Others, N (%)
Number (%)	136 (100.0%)	69 (50.7%)	32 (23.5%)	10 (7.4%)	11 (8.1%)	14 (10.3%)
ESMO 2016						
Low	78 (57.4%)	54 (78.3%)	22 (68.8%)	2 (20.0%)	0 (0.0%)	0 (0.0%)
Intermediate	12 (8.8%)	3 (4.3%)	4 (12.5%)	4 (40.0%)	1 (9.1%)	0 (0.0%)
High-intermediate	17 (12.5%)	10 (14.5%)	4 (12.5%)	2 (20.0%)	0 (0.0%)	1 (7.1%)
High	23 (16.9%)	2 (2.9%)	1 (3.1%)	2 (20.0%)	8 (72.7%)	10 (71.4%)
Advanced/metastatic	6 (4.4%)	0 (0.0%)	1 (3.1%)	0 (0.0%)	2 (18.2%)	3 (21.4%)
ESGO 2020						
Low	78 (57.4%)	52 (75.4%)	24 (75.0%)	2 (20.0%)	0 (0.0%)	0 (0.0%)
Intermediate	16 (11.8%)	5 (7.2%)	4 (12.5%)	4 (40.0%)	1 (9.1%)	2 (14.3%)
High-intermediate	14 (10.3%)	9 (13.0%)	2 (6.3%)	1 (10.0%)	0 (0.0%)	2 (14.3%)
High	23 (16.9%)	3 (4.3%)	1 (3.1%)	3 (30.0%)	8 (72.7%)	8 (57.1%)
Advanced/metastatic	5 (3.7%)	0 (0.0%)	1 (3.1%)	0 (0.0%)	2 (18.2%)	2 (14.3%)

EBRT, external beam radiotherapy; CCRT, concurrent chemoradiotherapy.

**Table 3 cancers-16-00965-t003:** Shift of cases between the ESMO 2016 and ESGO 2020 molecular risk groups.

	ESMO 2016					
ESGO 2020	Low	Intermediate	High-Intermediate	High	Advanced Metastatic	Total
**Low**	76 (97.4%)	0 (0.0%)	1 (1.3%)	1 (1.3%)	0 (0.0%)	78 (57.4%)
**Intermediate**	1 (6.3%)	12 (75.0%)	1 (6.3%)	2 (12.5%)	0 (0.0%)	16 (11.8%)
**High-intermediate**	0 (0.0%)	0 (0.0%)	11 (78.6%)	3 (21.4%)	0 (0.0%)	14 (10.3%)
**High**	1 (4.3%)	0 (0.0%)	4 (17.4%)	17 (73.9%)	1 (4.3%)	23 (16.9%)
**Advanced metastatic**	0 (0.0%)	0 (0.0%)	0 (0.0%)	0 (0.0%)	5 (100.0%)	5 (3.7%)
**Total**	78 (57.4%)	12 (8.8%)	17 (12.5%)	23 (16.9%)	6 (4.4%)	136 (100.0%)

## Data Availability

The authors confirm that all data underlying the findings described in this manuscript are fully available to all interested researchers upon request.

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
