# Peer review of "Assessing the New 2020 ESGO/ESTRO/ESP Endometrial Cancer Risk Molecular Categorization System for Predicting Survival and Recurrence"

_cancers, 2024, doi:10.3390/cancers16050965_

Round 1

Reviewer 1 Report

Comments and Suggestions for Authors

Considering that cases with POLEmut endometrial cancer have highly favorable outcome (>95% 5-year survival) despite aggressive pathologic features (lymphovascular space invasion and high-grade), please give some hard data explaining this outcome (related to 2020 ESGO...Molecular Categorization System). 

Reviewer 2 Report

Comments and Suggestions for Authors

The authors have conducted a fascinating study. The abstract and introduction effectively describe the necessity of this research.

However, in the methodology section, it is unclear which classification patients received treatment based on. I assume all patients were treated according to the ESMO 2016 classification, but it would be beneficial if the authors could confirm this.

The results are well-presented, and the discussions are both detailed and concise. Additionally, the references provided are up-to-date and pertinent.

Reviewer 3 Report

Comments and Suggestions for Authors

This retrospective study evaluated the efficacy of novel molecular classification in EC. In have the following comments:

1. What is the incidence of EC in Korea? This information should be added to the introduction section

2. Why did the authors choose time period between 2012-2021?

3. How many patients are treated for EC each year in the authors' institution and what were the exclusion criteria?

4. In the results section all cases should be reclassified into new 2023 FIGO classification and the results presented in appropriate table.

5. In the discussion section the authors should further discuss rare cases of EC with POLE and p53 mutations present. Is adjuvant therapy indicated in such cases in the early and advanced stages
